# THE EXTRAPOLATION POWER OF IMPLICIT MODELS

## ABSTRACT

Faced with out-of-distribution data, deep neural networks may break down, even on simple tasks. In this paper, we consider the extrapolation ability of *implicit* deep learning models, which allow layer depth flexibility and feedback in their computational graph. We compare the function extrapolation performance of implicit and non-implicit deep learning models on both mathematical tasks and real-world use cases in time series forecasting and earthquake location prediction. Throughout our experiments, we demonstrate a marked performance increase with implicit models. In addition, we observe that to achieve acceptable performance, the architectures of the non-implicit models must be carefully tailored to the task at hand. In contrast, implicit models do not require such task-specific architectural design, as they learn the model structure during training.

## 1 INTRODUCTION

Learning to extrapolate – the ability to estimate unknown values that extend the application of a method or conclusion beyond the observation range – is a core ability of human intelligence and an important development towards general machine intelligence. Although contemporary neural networks have demonstrated remarkable success in a myriad of domains, they struggle greatly when faced with data points outside of their training distribution (Barrett et al. (2018); Saxton et al. (2019)). In this work, we investigate how the data representation learned by implicitly-defined neural networks (Bai et al. (2019); Chen et al. (2018); El Ghaoui et al. (2021)) can help them perform function extrapolation on well-defined mathematical problems to real-world prediction tasks.

Implicitly-defined neural networks, such as implicit deep learning (El Ghaoui et al. (2021)) or deep equilibrium models (DEQ) (Bai et al. (2019)), are a general class of deep learning models that has been proposed as a potential alternative to classical neural networks. These implicit models do not operate on the premise of explicitly defined layers but instead have state vectors that are defined via an "equilibrium" (fixed-point) equation. Likewise, the outputs are determined implicitly by the same equation. Formally, for a given data point $u$, an implicit model solves the equilibrium equation $x = \phi(Ax + Bu)$, where $x$ is the equilibrium state for an input $u$, $\phi$ is a non-linear activation such as ReLU, and matrices $A, B$ are model parameters. The prediction is obtained by feeding the equilibrium state $x$ through an affine transformation, $\hat{y}(u) = Cx + Du$, where matrices $C, D$ are also model parameters. Recent results have shown the utility of implicit models (Bai et al. (2020); Gu et al. (2020); Tsai et al. (2022)). There has also been emerging work where the equilibrium state is interpreted as a closed-loop feedback system from a neuroscience perspective (Ma et al. (2022)). Instead of traversing nodes in a single direction (feed-forward) in the computational graph, inputs of implicit models can re-visit or reverse directions between nodes with closed-loop feedback. In this paper, we investigate whether implicit models possess better extrapolation capabilities, a core ability in mathematical reasoning, in comparison with specialized non-implicit models that are tailored to different problems. Our contributions are summarized in the following:

- **Evaluation on clean data.** We demonstrate the extrapolation power of implicit models on a series of mathematical problems whose data are generated from an underlying function.

- **Evaluation on noisy data.** Inspired by the success of the mathematical tasks, we further investigate the extrapolation performance of both implicit and non-implicit models on two real-world applications with noisy dataset.

- **Analysis.** We conduct analysis and ablation studies of *depth adaptability* and *close-loop feedback* and show that features learned by implicit models are more generalizable as compared to their non-implicit counterparts.

- **Architecture extraction.** We observe that implicit models learn task-based architectures during training, reducing the needs to carefully design the models prior.

## 1.1 RELATED WORK.

**Mathematical tasks.** Prior work to solve logical reasoning tasks has largely focused on developing specialized neural network models to learn algorithms (Graves et al.; Graves et al.; Kaiser and Sutskever; Liang et al.; Schwarzschild et al.). Neural Arithmetic Logic Units (NALU) explicitly represent mathematical relationships in their architecture (Trask et al. (2018)) for better arithmetic extrapolation. However, they later proved highly unstable to train (Schlör et al. (2020)). Nogueira et al. (2021) show transformers are well-suited for addition and subtraction tasks, achieving high accuracy on interpolation experiments. However, small transformers, BART (Wang et al. (2021)) and LLMs (Wei et al. (2022)) fail to reproduce functions when faced with numbers of larger magnitudes. On the contrary, when Charton (2022) tested transformers on matrix inversion and eigenvalue decomposition, they provided "roughly correct" solutions on out of distribution inputs, thus demonstrating some level of mathematical understanding. Transformers may be better suited for more complex mathematical problems.

**DEQs on OOD data.** The OOD generalization capabilities of deep equilibrium models (DEQs), a more generalized version of implicit models, have been demonstrated on Blurry MNIST, sequential tasks (Liang et al. (2021)), matrix inversion, and graph regression tasks (Anil et al. (2022)). Liang et al. and Anil et al. show that DEQs have a characteristic known as path independence; they converge to a similar fixed point regardless of initialization. Through this property, if DEQs iterate for longer before converging on OOD inputs, they could gain more information on the data and therefore outperform other models. We provide further evidence for this hypothesis with mathematical reasoning tasks. Taking into account the scaling difficulties of DEQs, we focus on small OOD tasks specifically in mathematical function extrapolation. In contrast with previous works, we perform extrapolation by directly modifying the training distributions, provide real-world forecasting applications of this phenomenon, and emphasize the importance of closed-loop feedback, in addition to the depth flexibility of DEQs, for our implicit models' success.

**Function extrapolation.** Xu et al. (2021) studied how ReLU MLPs and Graph Neural Networks extrapolate on quadratic, cosine and linear functions. They identify specific architectural choices that facilitate extrapolation such as encoding task-specific non-linearities in model features. Similarly in the vision domain, Webb et al. (2020) had introduced context normalization for more generalized features. In this paper, we argue that implicit models "adapt" to distribution changes therefore no longer requiring specific feature transformations for extrapolation. In a similar way, Wu et al. (2022) showcased how neural networks with Hadamard products (NNs-Hp) and polynomial networks (PNNs) possess advantages for arithmetic extrapolation. Although interested in similar tasks, this paper relies on MNIST digits rather than directly looking at mathematical functions. We are the first work to explore function extrapolation with implicit models.

## 1.2 BACKGROUND

Deep equilibrium models (DEQs) are defined by $f_w(u, x)$ with $f$ a function dependent on weights $w$, $u$ a data input and $x$ the hidden features. We solve for $x$ by repetitively applying $f$ to $u$ until we find a fixed point $x^*$ such that $x^* = f_w(u, x^*)$. In implicit models, a specific kind of DEQ first introduced by El Ghaoui et al., our function $f$ is defined as a non linear transformation based on four weights $A$, $B$, $C$ and $D$. They are learned during training and fixed at inference time. The implicit model takes a given input vector $u \in \mathbb{R}^p$ and produces a predicted output vector $\hat{y}(u) \in \mathbb{R}^q$ given by the following *implicit prediction rule*:

$$\hat{y}(u) = Cx + Du, \text{ where } x = \phi(Ax + Bu), \tag{1}$$

where $\phi$ is a given nonlinear map (the "activation function"), and the four matrices contain model parameters. Here, the so-called "state" vector $x \in \mathbb{R}^n$ contains the "hidden features" of the model; the latter cannot in general be expressed in closed-form, and is only implicitly defined via the so-called *equilibrium equation* $x = \phi(Ax + Bu)$. At a high level, we aim to "best" approximate our hidden features $x$ through a repeated non-linear transformation on our input $u$.

Throughout this paper, we consider a model has converged if $\|x_t - x_{t-1}\|_\infty < 3 \times 10^{-6}$ where $x_t$ and $x_{t-1}$ represent the current and previous implicit hidden states respectively. As demonstrated in El Ghaoui et al., implicit models will always converge under certain conditions (i.e. projecting the $A$ matrix onto an infinite norm ball) which we ensure in our experiments.

The implicit rule includes most current deep neural network architectures as special cases, from multi-layer perception (MLP) to long short-term memory networks (LSTM). Implicit models are a much wider class, as they have a lot more capacity, as measured by the number of parameters for a given dimension of the hidden features. To summarize, their representation differs in two key ways: the input is repetitively modified by the same weights (here: $A$, $B$, $C$ and $D$) and does not go through a fixed number of layers (as we instead stop the transformations when we converge). In this paper, we explore how these components could help function learning and extrapolation.

## 2 PROBLEM SETUP

We consider two types of implicit models throughout the paper: the standard implicit models as defined by (1) and the *implicitRNN*.

**ImplicitRNN.** Similar to a vanilla RNN, the implicitRNN processes the sequence input one by one. As shown in Figure 1, for each time step $i$, we are given an $i$-th element, $s_i \in \mathbb{R}^p$, in a sequence $(s_1, s_2, \cdots, s_t)$. The model input $u$ of implicitRNN in each time step is thus a concatenation of the $i$-th element $s_i$ and the previous hidden state $h_{i-1}$, identical to the vanilla RNN. The implicit prediction rule for implicitRNN is as follow:

$$h_0 = \mathbf{0}; \quad x_0 = \mathbf{0}; \quad u_i = \begin{pmatrix} s_i \\ h_{i-1} \end{pmatrix}$$
$$x = \phi(Ax + Bu_i) \quad \text{(equilibrium equation)}$$
$$\hat{y}_i(u_i) = Cx + Du_i \quad \text{(prediction equation)}$$
$$h_i = \hat{y}_i(u_i)$$

where the recurrent layer is replaced with an implicit one consisting of the equilibrium equation and prediction equation. We introduce the implicitRNN to compare implicit vs. explicit sequential models that both keep representations of data timestep by timestep.

### 2.1 EXTRAPOLATE ON MATHEMATICAL TASKS

Three types of functions are considered, from simple to complex: 1) identity function, 2) arithmetic operations, and 3) rolling functions over sequential data. To evaluate mathematical extrapolation, we have $u_{train} \sim P(U = u; \theta)$ and $u_{test} \sim P(U = u; \theta + \kappa)$ with $P$ a known distribution, $\kappa$ the distribution shift. All our code, models, dataset, and experiment setup are available at: github[1].

**Identity function.** He et al. (2016); Trask et al. (2018) show neural networks struggle to learn the basic task of identity mapping, $f(u) = u$, where models should return the exact input as given.

**Arithmetic operations.** We focus on two arithmetic operations: addition and subtraction. Replicating the task proposed by Trask et al. (2018), we randomly select four numbers $i, j, k, l$ from 1 to 50 where $i < j$ and $k < l$. For each sample, we construct two new numbers $a, b$ from the input $\vec{u} := \langle u_1, u_2, \cdots, u_{50} \rangle$ as follows: $a = \sum_{a=i}^{j} u_a$, $b = \sum_{b=k}^{l} u_b$. Finally, we predict $y = a + b$ for addition and $y = a - b$ for subtraction.

**Rolling functions.** We learn two rolling functions over a sequence: average and argmax. The rolling average task consists of predicting the average of the sequence up to the current time step $j$, for each timestep: $\sum_{i=1}^{j} u_i / j$. The rolling argmax task predicts the index of the maximum value seen by the model so far for each time step. We cast this to a classification problem by outputting a length $L = 10$ one-hot vector representing the index of the predicted maximum input seen so far, where $L$ is the length of the input sequence. We evaluate on predictions made to the final element.

---

[1]The link is redacted for anonymity purposes. We will release it in our camera-ready version

We compare implicit models with neural networks specially built to excel on each task (MLPs for simple functions, LSTMs by Hochreiter and Schmidhuber (1996) for sequential data and Google's Neural Arithmetic Logic Units (NALU) by Trask et al. (2018) for out-of-distribution (OOD) arithmetic) as well as with a variety of transformer architectures introduced by Vaswani et al. (2017). The details of each tasks and the model architectures are provided in Tables 3 and 4. We optimize all models using a grid-search and 5-folds cross-validation on in-distribution inputs. To ensure a fair comparison, we generate and train an implicit model with weights that have the same number of parameters as the DNNs' layers. Specifically for transformers, who tend to perform best on larger tasks, we reran our experiments with larger datasets, longer sequences and much larger models. Their performance didn't improve with size and so we chose not to include these results in our paper.

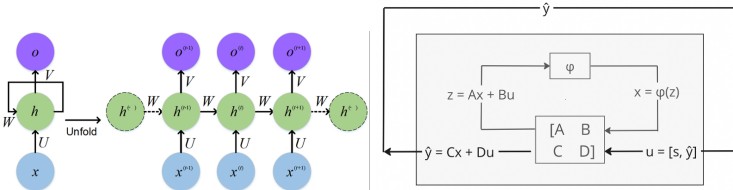

Figure 1: (**left**) Block diagram of a vanilla RNN (adapted from Feng et al. (2017)) and (**right**) implicit RNN. We replace the linear cell in a vanilla RNN with an implicit block not distinguishing between the output and the recurrent hidden state.

## 2.2 EXTRAPOLATE ON NOISY DATA

In addition to mathematical extrapolation where data is generated from a known underlying function, we consider extrapolation tasks in real-world problems where data are typically noisy with often unknown data generation function.

**Oscillating time series forecasting.** We first consider a simpler synthetic version called spiky time series forecasting where spikes obtained from a combination of sine functions are randomly inserted into a time series input (see A.3). Note, this is not an extrapolation task as the training and testing data have a similar distribution of spikes. To predict a real-life instance of sudden changes in time series, we also forecast AMC stock data volatility as it saw a drastic increase in average volatility at the start of 2021 (see Figure 7). Volatility is the expected amount by which a security's price might suddenly change. It measures the financial risk of an asset. We predict AMC's volatility over the next 10 minutes given its volume-weighted average trade price (VWAP) for each of the past 60 minutes. We calculate volatility as the variance of the VWAP prices[2] during the forecast period. Our training data goes from 2/01/2015 to 12/31/2020 and our validation set from 01/01/2021 to 12/31/2021. We do not make our data stationary through differencing or examining returns, since we specifically wish to evaluate our models on their ability to adapt to changes in price distribution. We compare implicit models to standard baseline models and linear regression (see Table 4).

**Earthquake location prediction.** The earthquake location prediction is a long-studied problem in seismology (Smith et al. (2021), Saad et al. (2021)). Given the observations of nearby stations recording seismic waves, we predict the location ($X$, $Y$ and $Z$) and p-wave travel time ($T$) of an earthquake. Accurately solving this problem before the destructive second wave(s) of an earthquake has huge humanitarian implications and also affects insurance underwriting, building codes, and policy writing. However, although proven effective in other seismology tasks, DNNs struggle in early warning system applications due to the sparsity of observations (Chuang et al. (2023)).

Following the methods presented by Chuang et al. (2023), we generate samples of seismic waves recorded by five stations with one designated anchor station used as a reference for all p-wave arrival times. Our input features are a station's coordinates ($x$, $y$, $z$), event-station back-azimuths ($\theta$), and the relative p-travel times ($p$) w.r.t. to the anchor station. We train on data synthetically generated to be between $90°E$ and $-90°E$, roughly corresponding to the Pacific Ring of Fire (see details about data generation in A.3). We then test on regions shifted from $10°E$ to $90°E$ beyond this Ring of Fire. We

---

[2]Normally, volatility is calculated as the standard deviation of returns. However, we wish to amplify the changes in distribution between our train and test set, so we use raw prices and the variance instead.

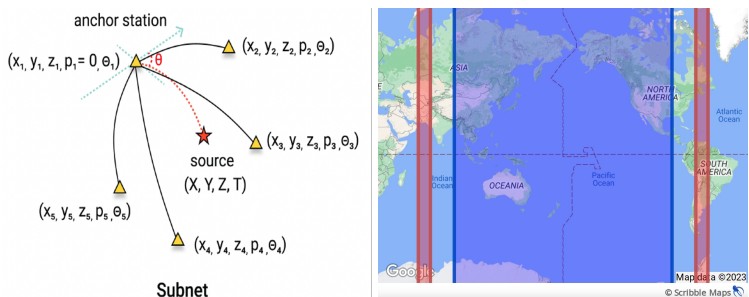

Figure 2: (**left**) Geometric visualization of one set of training features $(x_i, y_i, z_i, p_i, \theta_i)$ and its corresponding labels $(X, Y, Z, T)$. The triangles correspond to stations and the star corresponds to a source. (**right**) The map shows the training set region colored in blue, roughly corresponding to the Pacific Ring of Fire. The two red areas are the testing set regions for $k = 3$.

use a recently published deep learning model, EikoNet introduced by Smith et al., built specifically for earthquake location prediction, as a comparison baseline. Although earthquakes mostly occur in certain active tectonic boundaries, an extrapolated earthquake location prediction system can help detect natural or human-made earthquakes (caused by mining, oil and gas prediction for example) in new areas. In the case of explosion monitoring, universal mapping is extremely useful.

## 3    EXPERIMENTAL RESULTS

### 3.1    MATHEMATICAL EXTRAPOLATION

Figure 3 shows the test mean squared error (MSE) on the identify function task across various distribution shifts. Implicit model maintains the lowest test MSE ($< 5$) for test data that has a distribution shift from 0 to 25. For very large distribution shifts of up to 200 where $u_{\text{test}} \in \mathbb{R}^{10} \sim U(-205, 205)$, implicit model outperforms the transformer encoder by a factor of $10^5$ and the MLP by a factor of $10^3$. We observe that both MLP and transformer encoder continue to predict from the training distribution when tested on extrapolated data, thus increasing their error with the distribution shift. This simple task exposes the problem of over-fitting in non-implicit models.

Figure 4 show the results for the addition and subtraction arithmetic tasks. Implicit model outperform not only the various transformer encoders but also NALU, a model designed for math operations. As shown by Wei et al. (2022), transformers require much larger model sizes ($10^{23}$) to perform arithmetic. In contrast, an implicit model with only $7,650$ parameters successfully learns the operations, as it is able to replicate them on out-of-distribution data with a testing loss $< 1$ for shifts $< 100$. Throughout the experiment, we consistently observe that implicit models are well-adapted for logical tasks, especially with fewer training samples. Surprisingly, the specialized NALU model achieves the worst testing loss, $> 10^{10}$ for an extrapolation shift of only 10 as shown in Figure 4. Across our experiments, we weren't able to replicate robust out of distribution predictions with the NALU model.

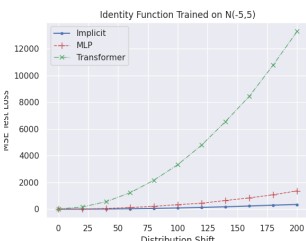
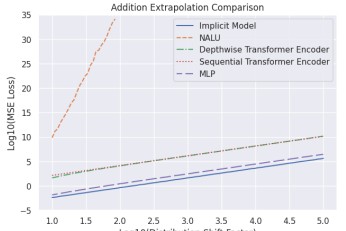
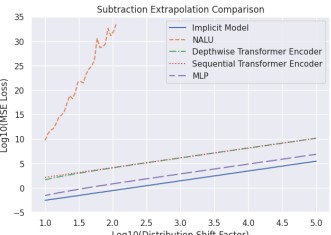

Figure 3: Test MSE for the identity function.

Figure 4: Test Log(MSE) for the arithmetic operations. The implicit model strongly outperforms all other models on OOD data.

As suggested by Liang et al. (2021), the more selective nature of implicit models may help them extrapolate better on logical tasks. For a specific input $u$, an implicit model's training only terminates

if we find a fixed point representation $x$ of $u$ through our equilibrium equation. During training on both arithmetic operations, we observed that our model failed to converge in under 300 iterations for at least 1/3 of the epochs. Implicit models thus appears to have the ability to filter out internal representations that do not help capture the underlying data generation function. On the other hand, MLPs' forward pass may overfit the training data, since it always terminates in a given number of steps: i. e., after the input has gone through all layers.

Lastly, Figure 5 show the extrapolation performance for the sequence modeling tasks. In the rolling average task (left plot of Figure 5), the LSTM and transformer behave very similarly, essentially replicating the training distribution whose rolling average is centered at 3. In contrast, the implicit model successfully extrapolates to higher values. For the rolling argmax, we use a full transformer with both encoder and decoder, where the target sequence is the right-shifted argmax labels. The transformer has an advantage over other models, as simply outputting the final element in this right-shifted sequence will provide the argmax of the input sequence up to but not including the current element. Given that argmax labels of a sequence are uniformly distributed, this will give an expected accuracy of 90% or $1 - 1/L$, where $L = 10$ is the sequence length. However, we observe that the standard implicit model and implicitRNN still outperform the LSTM and transformer variants. Moreover, the implicitRNN outperforms the standard implicit model. The rolling latent representation of the input sequence, therefore, seems beneficiary to argmax generalization. Comparing different transformer variants highlights the potential for small transformers to overfit to their positional encodings (PE) on simple tasks. Indeed, the transformer without PE, with no ability to distinguish between different positions in the sequence, outperforms the other two architectures: it hovers at the 90% benchmark, likely indicating that it correctly learned to always output the final value of its target sequence. As such, we demonstrate that in function extrapolation where the out-of-sample distribution could wildly differ from the in-sample distribution, the standard implicit model and implicitRNN achieve much better performance as compared to transformers and LSTMs, especially when an awareness of the relative positions between elements in a sequence is required. It is worth emphasizing that the implicitRNN has a similar structure as a vanilla RNN, simply with the standard recurrent cell swapped out for an implicit layer. Its significant performance increase over the LSTM suggests an exciting possibility that the implicit layer can learn proper memory gating procedures and avoid the vanishing gradient problems of RNNs via the $A$, $B$, $C$, $D$ weights that it shares across time steps. However, further experimentation, specifically with longer sequence lengths, would be required to verify this conjecture. More experimental results are included in Appendix A.2.

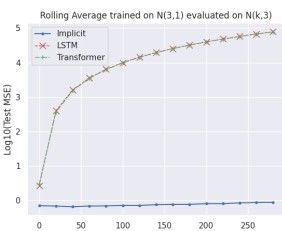 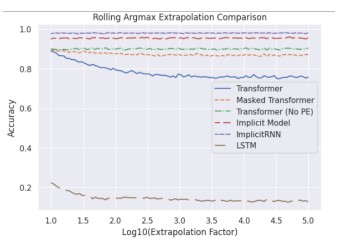 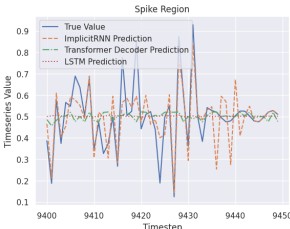

Figure 5: For rolling tasks, implicit models maintain close to constant loss ($\downarrow$) and accuracy ($\uparrow$) across shifts.

Figure 6: The implicitRNN most accurately models spike magnitudes.

## 3.2  Noisy Data Extrapolation

In our previous experiments, we applied implicit models to well-defined mathematical problems. We now aim to understand whether the extrapolation advantages of implicit models generalizes to real-world noisy data where we don't know the underlying function.

In the spiky time series forecasting, we want our models to capture the sudden and short-lived distribution changes in our generated data. During the extreme events that can occur in stock price, sales volume, and storage capacity prediction, many forecasting models fail due to their inability to extrapolate to these unprecedented patterns. Accurately adapting to the synthetic spikes in this task, therefore, has several real-life use cases. Table 1 shows that implicitRNN achieve three times lower test MSE on spiky time series task as compared to non-implicit models. In addition, Figure 6 highlights the capabilities of implicit model to capture the location and magnitudes of spikes. In

contrast, the transformer decoder and LSTM tend to output the average of the time series. It is worth mentioning that the tendency for the standard models to fall back on the mean also suggests their potential weakness in simulating time-series data. Motivated by these successes, we used implicit models to forecast the rapid increase in AMC stock volatility in 2021. We report the mean absolute percentage error[3] (MAPE) in Table 1. Due to the difficulty of the task, all models attain a MAPE greater than 1 but the implicit model outperforms them by a factor of at least 1.67.

Table 1: Train and test metrics (↓) in forecasting time series with sudden changes for synthetic (spiky time series) and real-world data (AMC stock volatility). Our architectures vary between tasks (see Table 4) but the implicit model outperforms all fine-tuned models.

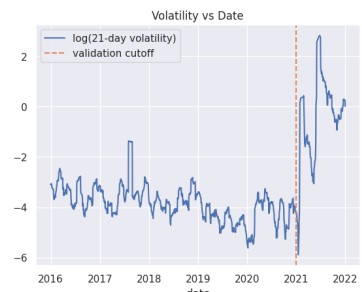

|  | Spiky Data (MSE) | | AMC Stock (MAPE) | |
| --- | --- | --- | --- | --- |
| **Model** | Train | Test | Train | Test |
| Transformer | 0.061 | 0.012 | 12.2 | 6.51 |
| ImplicitRNN | 0.015 | **0.004** | **2.61** | **1.71** |
| LSTM | **0.011** | 0.011 | 10.5 | 5.46 |
| MLP | - | - | 5.51 | 2.87 |
| Linear regression | - | - | 7.19 | 3.94 |

Figure 7: Time series of a 21-day rolling average of AMC stock volatility plotted on a log scale, highlights a drastic volatility increase at the beginning of our validation cutoff.

Finally, for the earthquake location prediction task, our implicit model improves the in-distribution test loss by $0.25\mathrm{e}{-3}$ over the EikoNet, designed for seismic data. More interestingly, for extrapolation with both models, we observe in the left panel of Figure 8 as $k$ increases, the implicit model performs increasingly better in terms of the extrapolated test set MSE. By the time $k = 2$, the implicit model has overtaken EikoNet, and when $k = 9$, the implicit model's test set loss is better than that of EikoNet by $1.59\mathrm{e}{-2}$. This translates to an average improvement of $11°$ longitude and $2°$ of latitude. Restricting the source latitude in addition to longitude during training could lead to a greater latitude extrapolation improvement. As seen in the right panel of Figure 8, the implicit model struggles a lot more with time and depth prediction, performing increasingly worse as $k$ increases. When $k = 9$, the implicit model performs 9.2 seconds and 409 km worse on average. Future work is needed to investigate whether training an implicit model exclusively on time and depth labels improves its performance. This two-pass approach is seen with traditional location prediction software such as HYPOINVERSE (USGS) since depth and time are much harder to constrain.

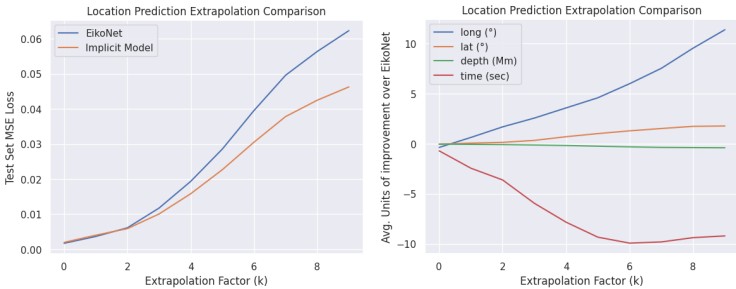

Figure 8: Extrapolation comparison between EikoNet and implicit model on the location prediction task as the extrapolation factor increases. The implicit model has the general edge in terms of MSE loss (**right**). Breaking down the prediction values (**left**) we observe that the implicit model only outperforms EikoNet in longitude and latitude predictions.

---

[3]The MAPE metric captures a model's ability to predict changes in magnitudes, relative to their size. As such, it can't be compared across distributions. Even though the models achieve higher RMSE on the validation data (for the implicit model: train RMSE = 0.001784, validation RMSE = 0.266366), the validation MAPE is lower since the average volatility is orders of magnitude larger in the validation set.

## 4 ANALYSIS

We identify two properties that enable implicit models to extrapolate well with relatively small datasets: "depth adaptability" (*i.e.*, it is not limited to a set number of layers) and "closed-loop feedback" (*i.e.*, inputs can re-visit the same node in a single pass through the model).

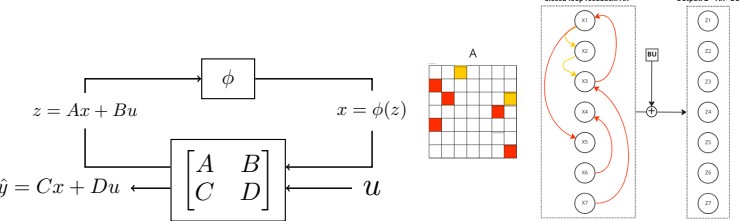

Figure 9: (**left**) Block diagram of an implicit model with input $u$, pre-activation state $z$ and our post-activation state $x$. Until convergence to a fixed point, $x$ is fed back into our model. (**right**) A matrix and node diagram representation of one iteration through the implicit model: $Ax + Bu$. In red, we have the weights that induce feedback in $A$, in yellow, the non-feedback weights.

**Depth adaptability.** A forward pass through an implicit model terminates when we either converge to a fixed point $X$ or fail to converge after a specified number of iterations as illustrated in the left diagram in Figure 9). We studied whether the input ($U$) complexity influences the number of iterations it takes for the model to reach an equilibrium. Across experiments, we observe that as our model learns parameter matrices, the number of iterations stabilizes for in-distribution inputs. On average, implicit models converged in approximately 15 iterations to learn the addition operation, 30 iterations for subtraction operation, and 175 iterations for rolling argmax. We can interpret the number of iterations performed on a given task as the model's perceived difficulty of that task. Following the results of Liang et al. (2021), we observed that as our input shifted farther away from the training distribution the number of iterations to converge augmented so $U$ went through more transformations by $A$, $B$, $C$ and $D$ (right two plots of Figure 10). As such, we interpret that implicit models dynamically "grow" in depth in order to adapt their feature space to extrapolated inputs by iterating longer as compared to in-distribution inputs. They benefit from low depth for in-distribution input (limiting overfitting) and higher depth for out of distribution input (limiting under fitting).

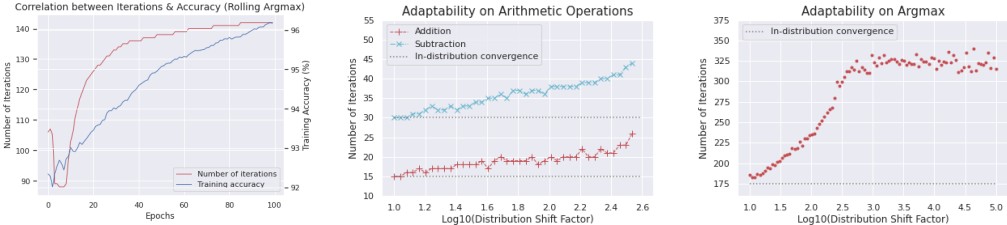

Figure 10: (**left**) Relationship between model performance during training and the number of iterations. Growth of implicit models as the input complexity increases in (**center**) arithmetic tasks and (**right**) rolling argmax.

**Closed-loop feedback.** Wiener (1948) first introduced "closed-loop feedback" as the ability of a system to correct itself based on its outputs. As explained by Patten and Odum (1981), the feedback is the portion of the output that is returned and partly determines the input at the next time step. Feedback from a neural network's outputs is encoded through the backward pass. As shown in the rightmost diagram of Figure 9, implicit models have built-in feedback over the course of one iteration due to the feedback connection in the parameter matrix $A$ (lower-triangular part). We observe that inputs do not only move forward from one set of nodes to the next, as in a standard feedforward neural network. Instead, they may re-visit the same nodes, or even move backward so the model can correct itself from one layer to the next whereas explicit models correct themselves after a passs through all layers. Ma et al. (2022) highlighted closed loop feedback as a strength of implicit models

and a similarity with human neural networks so we evaluated their utility in extrapolation through ablation studies on the lower triangular part of the model's $A$ matrix. We demonstrate through a $3 \times 3$ example that an implicit model with strictly upper triangular $A$ matrix has no feedback. We look at a single iteration at time $t + 1$, where $x^{t+1} = \phi(Ax^t + Bu)$. For simplicity, we can ignore the activation $\phi$ and $Bu$ terms, as our outputs from the previous iteration are encoded only in the $x^t$ term:

$$x^{t+1} \approx Ax^t = \begin{pmatrix} x_0^{t+1} \\ x_1^{t+1} \\ x_2^{t+1} \end{pmatrix} = \begin{pmatrix} \star & W_0 & W_1 \\ \star & \star & W_2 \\ \star & \star & \star \end{pmatrix} \begin{pmatrix} x_0^t \\ x_1^t \\ x_2^t \end{pmatrix} = \begin{pmatrix} \star \cdot x_0^t + W_0 x_1^t + W_1 x_2^t \\ \star \cdot x_0^t + \star \cdot x_1^t + W_2 x_2^t \\ \star \cdot x_0^t + \star \cdot x_1^t + \star \cdot x_2^t \end{pmatrix} \quad (2)$$

Closed-loop feedback corresponds to $x_i^t$ being used to generate $x_i^{t+1}$. We will show in equation 2, the $\star$ weights encode feedback. The $i$-th state at time step $t + 1$, $x_i^{t+1}$, depends on its past output $x_i^t$ directly through weights on the diagonal and indirectly through weights on the lower triangular matrix. As an example of this indirect dependence, $x_2^{t+1}$ depends on $x_1^t$; $x_1^t$ and $x_2^t$ both depend on $x_0^{t-1}$. Therefore, $x_2^{t+1}$ indirectly depends on $x_2^t$ with non zero lower triangular $\star$ weights.

We compare a regular implicit model (with feedback loops) and an ablated implicit model (without feedback loops) where the upper-triangularity of $A$ is enforced during training. The regular implicit models correspond to the ones used in our experiments, as described in Section 2. Table 2 shows the results of the math tasks for implicit models with and without feedback. The feedback loops help the models achieve significantly lower testing loss on inputs with distribution shifts. Figure 11 shows the ablation of both models across distribution shifts for the arithmetic and rolling sequential tasks. Ablation of the feedback harms model performance for subtraction but not for addition. The regular model has twice as many weights which may lead to overfitting on simpler tasks. Feedback loops seem to increase stability to distribution shifts for rolling average whereas it only provides better performance in calculating the rolling argmax of a sequence. We observe that aside from superior overall performance, the presence of closed-loop feedback also seems to make the model more impervious to distribution shifts: notably, we observe that its loss increases slower in the subtraction and rolling average experiments. Our analysis shows implicit models can adapt their architecture by learning the required depth, node connections, and correcting themselves based on past predictions in a single training iteration with closed-loop feedback.

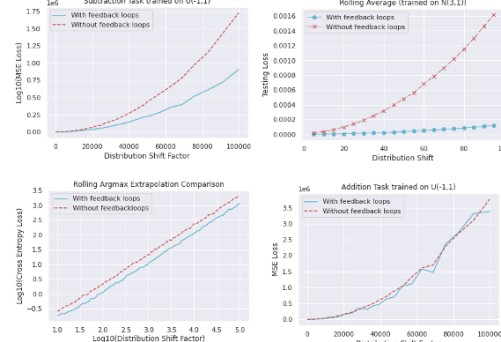

Table 2: Testing metrics (MSE ↓ for the arithmetic operations and rolling average, accuracy ↑ for rolling argmax) for a distribution shift of 100 comparing an implicit model trained with and without closed-loop feedback.

| Task | Feedback | No feedback |
|---|---|---|
| Add. | 3.06 | 4.07 |
| Sub. | 0.953 | 1.80 |
| R. Max. | 93.7% | 91.8% |
| R. Avg. | 1e−4 | 1e−3 |

Figure 11: Implicit models benefit from closed-loop feedback specifically in harder extrapolation tasks.

## 5 CONCLUSION

Implicit models improve upon the standard feed-forward layers of neural networks by relying instead on the convergence of an equilibrium equation to extract features from their input. In this paper, we demonstrate their capability of performing function extrapolation in mathematical tasks and on highly variable data. Throughout our experiments, implicit models strongly outperform non-implicit baselines on out-of-distribution inputs. The adaptive nature of training implicit models enables them to explore and identify optimal architectures with less hand-engineering, thereby providing a robust and scalable solution that offers strong performance on extrapolation tasks when faced with out-of-distribution data. These results motivate the further study of implicit models as a robust framework to excel under distribution shifts.

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

## A APPENDIX

### A.1 MODEL SPECIFICITES

We used ReLU activation function for our implicit models, transformers have a dropout of 0.1 and a layer norm epsilon of $\epsilon = 10^{-5}$. See Tables 3 and 4 on the next page for model architecture and experiment specificities.

Table 3: Details of training and out-of-distribution test set for each extrapolation task.

| Task | Training Distribution | Testing Distribution |
|------|----------------------|---------------------|
| Identity Function | $u_{\text{train}} \in \mathbb{R}^{10,000 \times 10} \sim U(-5, 5)$ | $u_{\text{test}} \in \mathbb{R}^{3,000 \times 10} \sim U(-\kappa, \kappa)$, where $\kappa$ ranges from 10 to 80 |
| Arithmetic Operations | $u_{\text{train}} \in \mathbb{R}^{10,000 \times 50} \sim U(-1, 1)$ | $u_{\text{test}} \in \mathbb{R}^{3,000 \times 50} \sim U(-\kappa/2, \kappa/2)$, $\kappa$ ranges from 10 to $10^5$ |
| Rolling Average | $u_{\text{train}} \in \mathbb{R}^{10,000 \times 10} \sim \mathcal{N}(3, 1)$ | $u_{\text{test}} \in \mathbb{R}^{3,000 \times 10} \sim \mathcal{N}(3 + \kappa, 1)$, $\kappa$ ranges from 5 to 100 |
| Rolling Argmax | $u_{\text{train}} \in \mathbb{R}^{10,000 \times 10} \sim U(0, 1)$ | $u_{\text{test}} \in \mathbb{R}^{3,000 \times 10} \sim U(0, \kappa)$, $\kappa$ ranges from $10^1$ to $10^5$ |
| Earthquake Location | 720,576 $(X, Y, Z)$ locations sampled between (90, -90)°E, 30 features | 20,016 samples in each extrapolation region $(90 - 10\kappa, 100 - 10\kappa) \cup (-100 + 10\kappa, -90 + 10k)$, $\kappa$ ranges from 1 to 9 |

### A.2 EXPERIMENTS MORE RESULTS

We provide more in-depth results on the OOD generalization capacities of implicit models for a specific small distribution shift in Figure 12. We compare the training and validation loss on the addition task of both implicit and MLP models where $u_{\text{train}} \in \mathbb{R}^{100} \sim U(1, 2)$ and $u_{\text{val}} \in \mathbb{R}^{100} \sim U(2, 5)$. Even with this small distribution shift, we observe a large improvement.

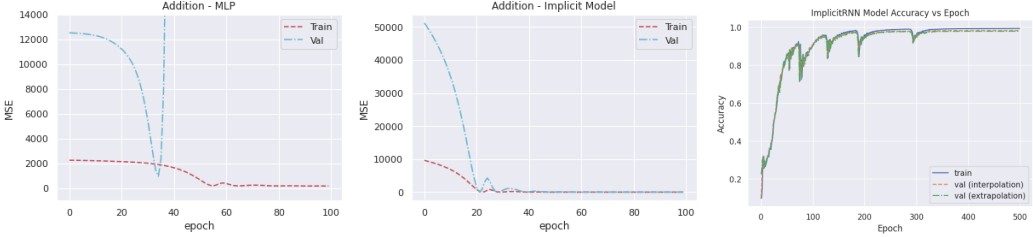

Figure 12: (**left** and **center**) The MLP test loss explodes whereas the implicit model achieves testing loss close to 0. (**right**) For rolling argmax prediction with extrapolation factor t = 10, our implicitRNN performs similarly on interpolated and extrapolated data.

### A.3 DATA GENERATION

**Spiky Data Generation**   Both the LSTM and the implicit model were trained on 7000 data points and tested on 3000 data points. The training regime featured 20 spiky regions of 100 data points each. The testing regime featured a proportionate amount of spiky regions. The data points in the spiky regions were sampled from $y = 5 \times (\sin(2x) + \sin(23x) + \sin(78x) + \sin(100x))$. We arbitrarily choose frequencies in $[0, 100]$ to generate a sufficiently spiky pattern. The magnitude of the spiky regions is at most 20. For the non-spiky regimes, the data points were sampled from $y = \sin(x)$ with added noise $\epsilon \sim \mathcal{N}(0, 0.25)$.

**Earthquake Data Generation**   To generate samples of seismic waves between specific longitudes, based on the methods presented by Chuang et al. (2023), we used a 1D velocity model called Ak135 from the Python library obspy.taup. Obspy is a Python framework used to process seismological

Table 4: Details of the explicit and implicit network architectures used in our experiments.

| Task | Baseline model | Implicit models | Transformers |
|---|---|---|---|
| Identity Function | MLP: $10 \times 9 \times 9 \times 10$ | Regular: $A \in \mathbb{R}^{4\times4}, B \in \mathbb{R}^{4\times10}, C \in \mathbb{R}^{10\times4}, D \in \mathbb{R}^{10\times10}$ | Encoder-decoder: $10 \times 10 \times 5$, 5 attention heads |
| Arithmetic Operations | • MLP: $50 \times 10 \times 10 \times 1$

• NALU: $50 \times 10 \times 10 \times 1$ | Regular: $A \in \mathbb{R}^{20\times20}, B \in \mathbb{R}^{20\times50}, C \in \mathbb{R}^{1\times20}, D \in \mathbb{R}^{1\times50}$ | • Sequential encoder: 1 layer, 10 attention heads, feedforward dim 50 - processes each array as a single sequence
• Depth-wise encoder: 1 layer, 1 attention head, feedforward dim 500, max PE length 50 - processes each element in a given array as a single sequence |
| Rolling Average | LSTM: $1\times18\times18\times1$ | Regular: $A \in \mathbb{R}^{32\times32}, B \in \mathbb{R}^{32\times10}, C \in \mathbb{R}^{10\times32}, D \in \mathbb{R}^{10\times10}$ | Encoder-decoder: $10 \times 10 \times 5 \times 10$, 5 attention heads |
| Rolling Argmax | LSTM: $1\times21\times21\times10$ | • Regular: $A \in \mathbb{R}^{36\times36}, B \in \mathbb{R}^{36\times10}, C \in \mathbb{R}^{10\times36}, D \in \mathbb{R}^{10\times10}$

• RNN: $A \in \mathbb{R}^{21\times21}, B \in \mathbb{R}^{21\times23}, C \in \mathbb{R}^{22\times21}, D \in \mathbb{R}^{22\times23}$ | • Masked encoder-decoder: 1 encoder layer, 1 decoder layer, 2 attention heads, feedforward dim 10, max PE length 10
• Unmasked encoder-decoder: 1 encoder layer, 1 decoder layer, 2 attention heads, feedforward dim 10, max PE length 10
• Unmasked encoder-decoder without PE: 1 encoder layer, 1 decoder layer, 2 attention heads, feedforward dim 10 |
| Spiky Time Series | LSTM: $1\times20\times20\times1$ | RNN: $A \in \mathbb{R}^{20\times20}, B \in \mathbb{R}^{20\times21}, C \in \mathbb{R}^{20\times20}, D \in \mathbb{R}^{20\times21}$ with a $20\times1$ linear layer | 1x10 linear layer (expansion) → masked decoder (1 layer, 2 attention heads, feedforward dim 2048, max PE length 10) → 10x1 linear layer (contraction) |
| Volatility Prediction | • LSTM: $1\times38\times38\times1$

• SGD Linear Regression

• MLP: $60 \times 50 \times 27 \times 27 \times 27 \times 10 \times 1$ | • Regular: $A \in \mathbb{R}^{53\times53}, B \in \mathbb{R}^{53\times60}, C \in \mathbb{R}^{1\times53}, D \in \mathbb{R}^{1\times60}$

• RNN: $A \in \mathbb{R}^{37\times37}, B \in \mathbb{R}^{37\times41}, C \in \mathbb{R}^{40\times37}, D \in \mathbb{R}^{40\times41}$ with a $40\times1$ linear layer | Sequential encoder (1 layer, 1 attention head, feedforward dim 2048, max PE length 60) → 60x1 linear layer |
| Earthquake Location Prediction | EikoNet: $270 \times 32 \times 128 \times 128 \times 128 \times 32 \times 4$ (42,500) | Regular: $A \in \mathbb{R}^{190\times190}, B \in \mathbb{R}^{190\times270}, C \in \mathbb{R}^{4\times190}, D \in \mathbb{R}^{4\times270}$ (42,680) | |

data. Kennett et al. (1995) demonstrate the accuracy of this model compared to real-world data (see specifically Figure 6). A 1D velocity model assumes the P-wave travel time (the duration the P-wave takes to travel from point A to point B) only depends on two attributes: the distance between the source and the receiver station and the depth of the source. We use this model to create a travel time

lookup table based on these two attributes. We then generate source locations from a mesh that spans the entire globe while adding perturbation to each latitude and longitude pair. We generate station locations using the source-station distances we have from the lookup table and place the stations in random orientations (azimuths) from the source.

