# OpenReview forum: "The Extrapolation Power of Implicit Models"
_ICLR.cc/2024/Conference — Submitted to ICLR 2024_

### Official Review · Reviewer_g5rm · 2023-10-31

**Soundness:** 2 fair
**Presentation:** 2 fair
**Contribution:** 2 fair
**Rating:** 3
**Confidence:** 4

**Summary:**

The paper demonstrates the ability of implicit models to perform function extrapolation and effectively handle highly variable data. The experiments show that implicit models outperform non-implicit models on out-of-distribution (OOD) data. The positive results suggest that further research into implicit models is warranted, as they offer a robust framework for addressing distribution shifts.

**Strengths:**

This paper demonstrates the extrapolation capabilities of implicit models by applying them to a series of mathematical problems with data generated from underlying functions. This study further explores how implicit models perform extrapolation on real-world applications with noisy datasets, comparing their performance to non-implicit models. Both ablation studies and an analysis are included to highlight the adaptability of implicit models, the importance of close-loop feedback, and how features learned by implicit models are more generalizable compared to their non-implicit counterparts. This paper observes that implicit models learn task-specific architectures during training, reducing the need for meticulous model design in advance. This adaptive feature is a significant contribution to handling various tasks effectively.

**Weaknesses:**

1 This paper studies the benefits of implicit models in terms of their extrapolation capabilities. However, it primarily describes this empirical finding and lacks a convincing analysis of its underlying causes. Specifically, this paper argues that the strong extrapolation capabilities of implicit models can mainly be attributed to two factors: the ability to adapt to varying depths and the inclusion of feedback in their computational graph. Nevertheless, the exact relationship between these two factors and their influence on extrapolation ability remains unclear. Further clarification on this matter is needed.

2 This paper conducts experiments on both mathematical tasks and real-world applications, including time series forecasting and earthquake location prediction, which is quite intriguing. However, the absence of experiments on benchmark datasets somewhat reduces the persuasiveness of the findings.

**Questions:**

Please see weaknesses.

---

### Official Review · Reviewer_qMje · 2023-10-31

**Soundness:** 2 fair
**Presentation:** 2 fair
**Contribution:** 2 fair
**Rating:** 3
**Confidence:** 4

**Summary:**

The paper aims to study the extrapolation power of Implicit Neural Networks. The authors use equilibrium models and a proposed implicit RNN model to perform evaluations on time series data -- both in the noisy and clean regime.

**Strengths:**

- The authors study various applications to show that implicit models indeed have superior extrapolation power.
- Analysis based on the closed loop feedback is a novel analysis that I haven't read earlier.

**Weaknesses:**

- _Architecture Extraction_ and _depth adaptability_ is not a novel contribution. Several publications on implicit modeling exploit this feature and have written about it. [1] [2] [3]
- The main contribution of the paper is not clear -- is it just showing implicit models are better in extrapolating in some tasks? In this case, the title of the paper should be revised. Since multiple papers have showed that implicit models have better extrapolation properties [4].
- ImplicitRNN is not a novel design as well -- it is simply a Neural ODE with backward Euler.

[1] https://papers.nips.cc/paper_files/paper/2018/file/69386f6bb1dfed68692a24c8686939b9-Paper.pdf

[2] https://proceedings.mlr.press/v139/pal21a.html

[3] https://proceedings.neurips.cc/paper/2020/file/2e255d2d6bf9bb33030246d31f1a79ca-Paper.pdf

[4] https://arxiv.org/abs/2001.04385

**Questions:**

1. Can the authors clarify on what their exact contributions are? The list in the paper is incorrect since other papers have demonstrated those capabilities before
2. Neural ODEs typically don't demonstrate stiffness, so the implicitRNN can be replaced by a neural ode with a better solver say RK45 / Tsit5 / VCAB3 and we should see similar performance. Also that some be more efficient and doesn't warrant a formulation like the one presented in the paper.

---

### Official Review · Reviewer_QgPM · 2023-11-06

**Soundness:** 3 good
**Presentation:** 3 good
**Contribution:** 2 fair
**Rating:** 5
**Confidence:** 3

**Summary:**

The authors work proposes an intensive study of the capacities of implicit models, in which the hidden representation is defined by a fixed point equation.
This work is mostly experimental.

In a first part, the authors analyze the relative performances of implicit models on a set of “noise-free” tasks (rolling mean, argmax, identity …) in which they show both the superior performance of implict models and their robustness to out-of-distribution data.

In the second part of the work addresses learning on real world datasets such as time series or earthquake source location. Finally, the authors provide an analysis of implicit models in terms of feedback loops and implicit depth.

**Strengths:**

The authors work is a convincing experimental study showing the merits of implicit models on various tasks encompassing both regression and classification (max).

It is presented in a clear way.

**Weaknesses:**

Since I am not very familiar with implicit models, a reminder on how implicit models are trained and / or  how and why they might converge, would have been a nice addition to the paper.

Moreover, most training details are not presented in the main text.

Finally, I found rather difficult to evaluate the last section of the authors work.

**Questions:**

1. Can the authors comment on their choice of experiments for both noise-free data / noisy - data ? Why not evaluate on classical benchmark such as Imagenet  / CIFAR ?
2. Can the authors comment on the limit in terms of capacities of implicit models ?
3. Is it of interest to stack multiple implict layers ?
4. What kind of optimizer is used to train implicit models ?

---

### Meta-Review · Area_Chair_3tCB · 2023-12-05

**Metareview:**

The author's work proposes a study of the capacities of implicit models on out-of-distribution datasets. However, the paper's novelty and contributions are unconvincing. The choices of the datasets and empirical settings are also unclear, making their conclusions less convincing. Therefore, all reviewers and I tend to reject this paper.

**Justification For Why Not Higher Score:**

The novelty and analysis are not enough for ICLR. And the empirical results setting is less convincing to demonstrate their conclusion.

**Justification For Why Not Lower Score:**

N/A

---

### Decision · Program_Chairs · 2024-01-16

Reject